# Role of Resolvins in Inflammatory and Neuropathic Pain

**DOI:** 10.3390/ph16101366

**Published:** 2023-09-27

**Authors:** Jaeik Park, Jueun Roh, Jingying Pan, Yong Ho Kim, Chul-Kyu Park, Youn Yi Jo

**Affiliations:** 1Gachon Pain Center and Department of Physiology, Gachon University College of Medicine, Incheon 21999, Republic of Korea; jaeik0518@gachon.ac.kr (J.P.); jueun9392@gmail.com (J.R.); panjingying0505@126.com (J.P.); eruo16@gmail.com (Y.H.K.); 2Department of Histology and Embryology, Medical School of Nantong University, Nantong 226007, China; 3Department of Anesthesiology and Pain Medicine, Gil Medical Center, Gachon University, Incheon 21565, Republic of Korea

**Keywords:** resolvin, inflammatory pain, neuropathic pain, inflammation, pain management, dorsal root ganglia, spinal cord

## Abstract

Chronic pain is an unpleasant experience associated with actual or potential tissue damage. Inflammatory pain alerts the body to inflammation and promotes healing; however, unresolved inflammation can lead to chronic pain. Conversely, neuropathic pain, due to somatosensory damage, can be a disease in itself. However, inflammation plays a considerable role in the progression of both types of pain. Resolvins, derived from omega-3 fatty acids, actively suppress pro-inflammatory mediators and aid in the resolution of inflammation. Resolvins alleviate various inflammatory and neuropathic pain models by reducing hypersensitivity and regulating inflammatory cytokines and glial activation in the spinal cord and dorsal root ganglia. Thus, resolvins are a promising alternative for pain management with the potential to reduce the side effects associated with conventional medications. Continued research is crucial to unlock the therapeutic potential of resolvins and integrate them into effective clinical pain management strategies. This review aimed to evaluate the literature surrounding the resolvins in inflammatory and neuropathic pain.

## 1. Introduction

Pain is an unpleasant sensory and emotional experience associated with actual or potential tissue damage or described in terms of such damage [1]. Nociceptive pain is mediated by nociceptors in response to specific stimuli, alerting the body to potential harm, avoiding harmful stimuli, and protecting the body [2].

Inflammation triggers inflammatory pain and contributes to certain neuropathic pain conditions. Inflammatory pain functions as a defense system by identifying inflammation via symptoms, such as swelling, redness, and heat; however, it also plays a role in the healing process of the affected area [3,4]. Chronic inflammation may occur in cases of unresolved inflammation, resulting in chronic pain [5]. Unlike nociceptive and inflammatory pain that arises from disease or tissue damage, neuropathic pain can be a disease in itself [6,7]. It is generally chronic and is caused by lesions within the somatosensory system, which bidirectionally communicates with the immune system [8,9,10,11]. Single-cell RNA sequencing has revealed upregulated immune system-related genes in the dorsal root ganglia (DRG), sciatic nerves, and spinal cord in a neuropathic pain model [12,13,14]. Consequently, regulating inflammatory responses under pathological conditions offers potential pain alleviation, and inflammation-modulating medications are promising therapeutic agents for pain management [15,16,17,18]. Therefore, there has been tremendous interest in studying drugs targeting inflammation and exploring the underlying mechanisms involved in regulating inflammation in diverse pain models.

In recent decades, studies on the analgesic effect of natural compounds or natural compound-based endogenous substances have been actively conducted. Typically, marine natural products (MNPs) derived from marine life distributed in the oceans, which comprise two-thirds of the earth’s surface, are examples. Omega-3 polyunsaturated fatty acids (ω-3 PUFAs), which are precursors of resolvins (Rvs), are representative MNPs from fish oils [19,20]. Moreover, ω-3 PUFAs are precursors of several types of specialized pro-resolving mediators (SPMs) that contribute to anti-inflammatory action, such as protectins, maresins, and Rvs [19]. A recent report indicates that MNPs with analgesic effects include axinelline A and its analogs from *Streptomyces axinellae* at the bacterial level [21], as well as aaptamine from sea sponges [22]. In addition, more than 10,000 types of MNPs from algae, bryozoans, fungi, and other marine organisms have shown antiviral, antioxidant, anti-inflammatory, and anticancer properties, and it is expected that such bioactivity can exhibit effective analgesic action [19]. Outside the ocean, phytoconstituents, which are non-nutrient plant chemical or bioactive compounds, are also representative natural compounds [23]. Traditionally, *Cannabis sativa* (cannabis) and its extracts have been used for centuries to treat several diseases, including chronic pain [24,25,26]. Among the various components of cannabis, reports note that cannabidiol and tetrahydrocannabinol are mainly involved in various bioactivities by interacting with various orphan G protein-coupled receptors (GPCRs) and nuclear factors or regulating several ionotropic receptors, transporters, and enzyme activity [27,28]. In addition, until recently, research to improve the efficiency of cannabidiol, which shows irregular or low-efficiency effects, has also been reported [29,30,31]. Widely known drugs such as opioids and morphine are also phytoconstituents obtained from *Papaver somniferum* (opium poppy) [32,33], and the bioactivities and mechanisms of various phytoconstituents are being studied. Further research is required regarding the mechanisms of action of these natural compounds to allow their increased use for more effective treatment. 

Chronic pain can occur as joint pain, muscle pain, and headaches due to various causes, such as trauma or disease. Sufficient research on the mechanisms and properties of these natural compounds may allow for appropriate treatment with high efficiency and few side effects in each administration. For this reason, this review summarizes the studies on the anti-inflammatory and analgesic effects of Rvs, one of the SPMs synthesized from ω-3 PUFAs.

## 2. Biosynthesis of Resolvins

Omega-3 fatty acids are primarily acquired through dietary sources, encompassing fish, fish oils, seafood, and some plants [34,35]. Because it cannot be endogenously synthesized within the human body, omega-3 must be acquired through dietary intake [36]. These fatty acids serve as the substrate for synthesizing Rvs, lipid mediators derived from omega-3 fatty acids, specifically eicosapentaenoic acid (EPA) and docosahexaenoic acid (DHA). Rv biosynthesis has been extensively studied both in vivo and in vitro. Various immune cells, including macrophages, eosinophils, neutrophils, and endothelial cells, are involved in this process by releasing specific fatty acids and utilizing enzymes to convert DHA and EPA into precursor molecules for Rv D (RvD) and Rv E (RvE) series, respectively [37,38]. 

### 2.1. Biosynthesis of D-Series Resolvins

DHA is metabolized by 15-lipoxygenase (15-LOX) into 17(S)-hydroperoxy-docosahexaenoic acid (17(S)-Hp-DHA) and catalyzed by peroxidase into 17(S)-hydroxydocosahexaenoic acid (17(S)-HDHA) [39,40]. Next, 17(S)-HDHA undergoes enzymatic oxidation by 5-LOX, leading to the formation of two hydroperoxyl intermediates: 7(S)-hydroperoxy-17(S)-hydroxydocosahexaenoic acid (7(S)-Hp-17(S)-HDHA) and 4(S)-hydroperoxy-17(S)-hydroxydocosahexaenoic acid (4(S)-Hp-17(S)-HDHA) [40,41], which are epoxygenated to form 7(8)-epoxy-17(S)-HDHA and 4(S)-epoxy-17(S)-HDHA, respectively [40,42]. Finally, 7(8)-epoxy-17(S)-HDHA and 4(S)-epoxy-17(S)-HDHA are hydrolyzed by leukotriene A4 (LTA4) using Zn^2+^ as a cofactor, yielding two types of RvD family members [40,43,44]. RvD1 and RvD2 are synthesized from 7(8)-epoxy-17(S)-HDHA [41,45,46], whereas RvD3 and RvD4 are synthesized from 4(S)-epoxy-17(S)-HDHA [42,47,48,49]. The two hydroperoxyl intermediates can also be transformed with hydroperoxyl reductase into other types of D-series Rvs; 7(S)-Hp-17(S)-HDHA can be converted into RvD5, and 4(S)-Hp-17(S)-HDHA can be converted into RvD6 [41,42,50] (Figure 1).

Additionally, synthesis of the RvDs can be triggered with aspirin and cyclooxygenase-2 (COX-2) [51]. This reaction is initiated with the conversion of DHA into 17(R)-Hp-DHA catalyzed by aspirin-dependent acetylated COX-2. 17(R)-Hp-DHA formed in this way is a stereoisomer of 17(S)-Hp-DHA but similar to the synthesis process of RvDs [51,52]. It is catalyzed by peroxidase, oxidized by LOX5, hydrolyzed by LTA4H, and converted into aspirin-triggered RvDs (AT-RvDs) through stereoisoform intermediate compounds. R-type intermediate compounds differ from the S-types in the position of the OH group structurally [51]. In terms of physiological characteristics, R-types have a longer biological half-life and a higher resistance to metabolic inactivation by oxidoreductase than S-types [51,52,53]. Contrary to the characteristics of these intermediate compounds, AT-RvDs have an equal level of bioactivity with RvDs. For example, studies of Sjogren’s syndrome, a chronic inflammatory autoimmune disease [54,55], showed that symptom relief and tissue recovery by RvD1 and AT-RvD1 were similar [56,57]. RvD3 and AT-RvD3 induce pro-resolving activity by regulating immune cell activation [47]. This is supported by the resolution of the inflammatory response and tissue recovery by inhibiting the NF-κB pathway of RvD3 and AT-RvD3 in acute lung injury and acute respiratory distress syndrome caused by barrier dysfunction of normal endothelial and epithelial cells [58,59]. Furthermore, this review comments on the analgesic effects of AT-RvD1 identified in inflammatory and neuropathic pain models.

### 2.2. Biosynthesis of E-Series Resolvins

EPA is converted by acetylated COX-2 or microbial cytochrome P450 into 18-hydroxyeicosapentaenoic acid (18-HEPE) [60,61]. Subsequently, 18-HEPE is converted into RvE2 by peroxidase or epoxidated to generate 5S-hydroperoxy-18-HEPE and 5S,6S-epoxy-18R-HEPE via 5-LOX. These are then hydrolyzed to yield RvE1 by LTA4 hydrolase [60,61,62]. Additionally, oxygenation of EPA by 5-LOX leads to either RvE3 or 15S-H(p)EPE formation, which can be further transformed by 15-LOX or 5-LOX into 15S-hydroxy-5S-H(p)EPE and eventually converted into RvE4 [60,61] (Figure 2).

## 3. Resolvins as Potential Therapeutic Targets in Inflammation

Rvs play a critical role in resolving various inflammation-related physiological responses. Recently, Rvs have emerged as promising therapeutic agents for pain management by specifically targeting inflammatory processes. Many studies have demonstrated the role of Rvs in inflammation. For instance, circulating levels of Rvs are significantly reduced in some inflammation-related diseases, such as aneurysmal subarachnoid hemorrhage, acute myocardial infarction, bipolar disorder, irritable bowel syndrome, acromegaly, and Hashimoto’s thyroiditis [50,63,64,65,66,67]. 

Regarding the underlying mechanisms, Rvs actively suppress the production of pro-inflammatory mediators, such as cytokines and chemokines, thereby exerting anti-inflammatory effects. In an in vitro study using peripheral nerve-derived stem cells, RvD1 reduced pro-inflammatory cytokines and increased anti-inflammatory cytokines when peripheral nerve-derived stem cells were treated with lipopolysaccharide (LPS) [68]. Furthermore, RvD1 promoted the regeneration of cavitation and tissue contraction in an SCI model [68], resolved neuroinflammation, reduced interferon (IFN)-γ levels in the cerebrospinal fluid in Parkinson’s disease, and attenuated the gene expression and release of interleukin (IL)-1β and IL-18 after ethanol and LPS treatment in bone marrow-derived macrophages [69,70]. RvD2 did not affect the increase in tumor necrosis factor (TNF)-α, IL-6, IFN-γ, IFN-β, IL-1β, and monocyte chemoattractant protein-1 levels in the plasma of cecal ligation and a puncture-induced infectious sepsis model [71]. However, RvD2 significantly reduced the lung lavage levels of IL-23 in lung infections caused by *Pseudomonas aeruginosa* [71]. RvD3 restored decreased IL-10 and increased IL-6 and TNF-α expressions in LPS-treated RAW264.7 macrophages [72]. Moreover, RvE1 and RvE2 increased IL-10 mRNA expression and IL-10Rβ protein expression in human monocyte U937 cells, whereas RvE3 had slight or no significant effects on IL-10 mRNA expression and IL-10α receptor levels [73] (Figure 3).

Rvs also enhance phagocytic function in macrophages [73,74,75]. D-series Rvs regulate phospholipase D, a membrane lipase that modulates phagocytic function in M1 and M2 macrophages [75]. RvD2 and RvD3 enhanced bacterial phagocytosis in human macrophages by approximately 80% more than the controls [76]. RvD2 also significantly increased non-inflammatory alveolar macrophages, which enhanced bacterial clearance through phagocytosis in lung infections caused by Pseudomonas aeruginosa [71]. Additionally, RvD3 and RvD4 significantly decreased granuloma formation in peripheral blood mononuclear cells [77]. Deoxy derivatives of RvE3 suppressed the collection of peritoneal exudate cells in bacteria-induced peritonitis, indicating that deoxy derivatives of RvE3 have anti-inflammatory activities [78] (Figure 3).

Rvs promote the switch in the phenotype of macrophages into M2 and inflammatory cells’ clearance from the site of inflammation, facilitating inflammatory resolution and contributing to the restoration of tissue homeostasis [44,60,75,79]. RvD1 enhances efferocytosis by inhibiting aging-induced Mer Tyrosine Kinase cleavage in macrophages [80]. RvD2 treatment increases the proportion of F4/80+ macrophages expressing anti-inflammatory macrophage markers (CD206, Arginase-1, and CD163) and significantly decreases macrophages expressing inducible nitric oxide synthase (an M1 macrophage marker) [81]. Furthermore, RvD3 enhances the phagocytosis of apoptotic neutrophils by human macrophages in the low dose range (pM to low nM) [76]. Synthetic RvE4 showed a concentration-dependent increase in the macrophage efferocytosis of senescent red blood cells. The concentrations of RvE4 that showed the most significant enhancement were 1 nM and 10 nM, and the estimated half-maximal response (EC50) was approximately 0.29 nM [60] (Figure 3).

In summary, Rvs help to restore tissue homeostasis and contribute to the overall resolution of inflammation by inhibiting immune cell activation and reducing the release of pro-inflammatory mediators. Preclinical and clinical studies have demonstrated the efficacy of Rvs in mitigating pain associated with various conditions. Thus, there is promise in alleviating pain in animal models of inflammatory and neuropathic pain and other chronic pain conditions by targeting the underlying inflammatory processes. Furthermore, Rvs enhance the effectiveness of conventional analgesic therapies, suggesting potential as adjunct treatments for pain management.

## 4. Functions of Resolvins in Inflammatory Pain

Inflammatory pain increases sensitivity due to the inflammatory response associated with tissue damage. Traditionally, nonsteroidal anti-inflammatory drugs (NSAIDs), such as cyclooxygenase inhibitors, have been used to manage inflammatory pain [82,83]. However, the side effects, including gastrointestinal, cardiovascular, hepatic, renal, cerebral, and pulmonary complications, have been reported in multiple placebo-controlled trials and meta-analyses studies [83]. In contrast, only minor side effects resulting from prolonged systemic intake of ω-3 PUFAs have been reported, primarily encompassing gastrointestinal distress, platelet aggregation, and the immune response to infection. To the best of our knowledge, no systemic side effects have been reported with the use of Rvs [84,85,86,87,88,89]. Therefore, the development of effective and safe treatments for inflammatory pain has been studied using various inflammatory pain models. 

An inflammatory pain model can be established with an intraplantar injection of a nociceptor agonist, such as capsaicin (transient receptor potential vanilloid-1 (TRPV1) agonist) [90] or inflammatory cytokines. This can be confirmed by measuring changes in pain-like behavior, such as spontaneous pain (appearing in actions such as licking, shaking, and biting), mechanical hyperalgesia or allodynia, and thermal hypersensitivity [82] (Table 1).

### 4.1. Formalin-Induced Pain Model

A formalin-induced pain model can be established by the subcutaneous injection of formalin solution (2.5–5%) into rodents’ hind paws [97]. After injection, the measurements comprise two separate phases with different durations and underlying mechanisms. Phase I, the initial acute phase, is mediated by direct activation of the transient receptor potential ankyrin 1 (TRPA1) channel; Phase II is an inflammatory and central nociceptive sensation [98]. This strategy is one of the most widely studied inflammatory pain models in Rv research. Intraplantar injection of low-dose RvD1 attenuates both phases of formalin-induced spontaneous pain behavior time [94]. However, intrathecal administration of other Rvs (RvE1 [92] and RvD2 [93]) attenuates only the pain behavior time of Phase I. These studies suggest that peripheral administration of Rvs is involved in the regulation of TRPA1 channels and the inhibition of inflammatory responses but does not directly inhibit TRPA1 in inflamed tissue by intrathecal injection. Interestingly, intrathecal injection of RvD5 attenuates only the pain behavior time of Phase II in males [91]. 

### 4.2. Complete Freund’s Adjuvant (CFA)-Induced Pain Model

CFA is a suspension of desiccated mycobacteria in paraffin oil and mannide monooleate that induces inflammation, tissue necrosis, and ulceration [99]. Intraplantar injection of CFA elicits week-long inflammatory pain [92]. However, an attenuation of mechanical allodynia and thermal hyperalgesia was reported in an RvD1 study [94]. Another study focused on the Rv administration point (RvD1 and RvE1 intrathecally), which may affect the persistence of the analgesic effect [92]. At concentrations higher than those of RvD1 and RvE1, intrathecal injection of RvD2 attenuated heat hyperalgesia [93]. Intraperitoneal injection of aspirin-triggered (AT) RvD1 also attenuated mechanical allodynia [95]. Moreover, pro-inflammatory cytokines, such as TNF-α and IL-1β, were decreased by AT-RvD1 in the ipsilateral rat hind paw [95]. 

### 4.3. Capsaicin-Induced Pain Model

Capsaicin selectively activates TRPV1, which is enriched in nociceptive primary afferent neurons [90]. A relatively short-lasting spontaneous pain model can be established by the intraplantar injection of capsaicin [100]. Although less pronounced than in previous models, some Rvs, such as RvD2 [93] (intraplantar) and RvE1 [92,93] (intraplantar and intrathecal), attenuated spontaneous pain.

### 4.4. Carrageenan-Induced Pain Model

Carrageenan-induced acute and local inflammation is one of the most popular tests used to screen for anti-inflammatory activity [101]. Mechanical and thermal hypersensitivity increased on the ipsilateral side of the carrageenan-injected mouse’s hind paw [92,93]. Attenuation of mechanical and thermal hyperalgesia by intraplantar injection of RvD2 has been reported [93]. A study on the anti-inflammatory effect of RvD1 and RvE1 confirmed the attenuation of paw edema and inflammatory cytokine expression in the ipsilateral hind paw [92].

### 4.5. Allyl Isothiocyanate (AITC)-Induced Pain Model

AITC, a major component of natural mustard oil, has only been investigated in a neurogenic inflammatory model, as it is a natural agonist of TRPA1 [102]. Intraplantar injection of AITC induces spontaneous pain in experimental animals. The intrathecal injection of RvD1 or RvD2 attenuated the AITC-induced spontaneous pain behavior time [93].

### 4.6. Cytokine-Induced Pain Model

Pro-inflammatory cytokines, such as IL-1β, IL-6, and TNF-α secreted by activated macrophages, are involved in the pain process [103]. Several studies have identified reductions in pro-inflammatory cytokines by Rvs in neuropathic and inflammatory pain models. In addition, the intrathecal injection of recombinant pro-inflammatory cytokines (rIL-17 [96] or TNF-α [92]) evoked mechanical and thermal hyperalgesia; this pain was reversed with Rvs [92,96].

## 5. Functions of Resolvins in Neuropathic Pain

Neuropathic pain is caused by damage or disease affecting the somatosensory system [104]. The pain phenotype is determined with multiple alterations from the ectopic generation of an action potential to neuroimmune interactions [6]. Due to the varied etiology, neuropathic pain treatments are being developed in various ways, including the control of nociceptive channels and neuroinflammation. The Rv series, a specialized pro-resolving mediator, has an analgesic effect in several neuropathic pain models, which can be established in two main ways: tissue or nerve injury [100]. Additionally, unlike the inflammatory pain models, Rv studies using neuropathic pain models have confirmed changes in inflammatory factors and regulation of glial activation at the levels of the DRG and spinal cord (Table 2).

### 5.1. Chemotherapy-Induced Peripheral Neuropathy (CIPN) Pain Model

CIPN is a major complication of chemotherapies [108], such as paclitaxel (taxanes) and oxaliplatin [109]. Periodic intraperitoneal paclitaxel injections induced hind paw mechanical and thermal hypersensitivity [110]. CIPN is caused by glial cell activity and inflammatory cytokine and chemokine regulation [111]. Therefore, several studies have demonstrated the analgesic effects of Rv series using the CIPN mouse model. Intraperitoneal injection of RvD1 on day 1 after paclitaxel injection attenuated mechanical allodynia and thermal hyperalgesia in response to heat and cold stimuli [105]. Moreover, RvD1 reduced M1 macrophage activity and increased IL-10 expression and anti-inflammatory cytokines in the DRG and sciatic nerve [105]. Sexual dimorphism of RvD5 was reported in the mechanical allodynia regulation of TRPV1 or TRPA1 knock-out CIPN mice [105].

### 5.2. Chronic Constrictive Injury (CCI) Mouse Model

The CCI pain model is a chronic neuropathic pain model that loosely ligatures the sciatic nerve three to four times [112]. This surgical neuropathic pain model was developed to mimic peripheral nerve injury and exhibit mechanical allodynia and thermal hyperalgesia [113]. Post-surgical administration of RvD2 at low doses (500 ng) via the intrathecal route and higher doses (5 μg) via the intravenous route significantly attenuated mechanical allodynia and thermal hypersensitivity [105]. In addition, Rv-D2 reduced IL-17 and CXCL1 expression and astrocyte activity in the spinal cord dorsal horn [105]. A pre-surgical administration of RvE1 at a lower dose (100 ng, 3 days before surgery) also attenuated mechanical allodynia in the CCI mouse model [106]. Furthermore, RvE1 reduced TNF-α expression and microglia and astrocyte activity in the dorsal horn of the spinal cord [106].

### 5.3. Spinal Nerve Ligation (SNL) Rat Model

The SNL model is a neuropathic pain model developed from tight ligation of the L5 or L5 and L6 segmental spinal nerves in rats [114]. This neuropathic pain model has been widely used in studies on pain mechanisms and screening tests for analgesic candidates [114]. Intrathecal injection of RvE1 or AT-RvD1 attenuated mechanical and thermal hyperalgesia [106,107]. In addition, the activities of microglia and various types of pro-inflammatory cytokine expression, such as IL-1β, IL-18, and TNF-α, were reduced by AT-RvD1 [107].

## 6. Perspective

### 6.1. The Preparation of NSAIDs and Resolvins

NSAIDs are a class of pharmaceuticals widely employed for pain relief, inflammation reduction, and fever reduction [115]. NSAIDs typically feature organic compounds as their primary active ingredients, such as ibuprofen, aspirin (acetylsalicylic acid), naproxen, or diclofenac [116]. These compounds undergo various chemical reactions to synthesize the desired product [117,118,119]. Following the implementation of stringent quality control measures to ensure the active ingredient’s purity and consistency, it is then formulated into diverse pharmaceutical dosage forms, including tablets, capsules, liquid suspensions, or creams.

Rvs are typically biosynthesized from omega-3 fatty acids and commonly extracted from EPA and DHA. The isolation of EPA and DHA involves a meticulous purification process followed by their utilization as substrates in both enzymatic and chemical reactions to catalyze the synthesis of RvE and RvD, respectively (Figure 1 and Figure 2). The resultant Rv mixture undergoes purification and separation techniques, such as chromatography, to yield the targeted Rv compounds. As Rvs are presently in the preclinical stages, certain prerequisites and processes must be completed before embarking on drug formulation for patient use.

### 6.2. Challenges in Resolvins Production

The market size value of NSAIDs has consistently demonstrated an upward trend in recent years, whereas the market value of Rvs is currently indeterminable. The cost of a recombinant LOX-based chemosynthesis process is more than USD 15,000 per mg and is known to be significantly inefficient in terms of price compared to NSAIDs. Moreover, due to the properties of Rvs, such as photosensitivity, susceptibility to heat, and proneness to oxidation, the production costs are considerably elevated, rendering large-scale manufacturing and the establishment of a novel drug industry chain a formidable challenge [120].

### 6.3. Administration Routes for Resolvins

Intraplanar and intrathecal Rv injections have traditionally been used for administration routes [91,92,93,96,121]. Intravenous, intranasal, and oral Rv injections have modulated immune cells and alleviated symptoms of immune diseases [122,123,124,125,126]. In particular, the intranasal injection of Rvs is a non-invasive route for effective drug delivery to the brain [127]. Intranasal injection of Rvs elicited antidepressant-like effects through the release of brain-derived neurotrophic factor and vascular endothelial growth factor in the medial prefrontal cortex (mPFC) and hippocampal dentate gyrus, as well as the mammalian target of rapamycin complex 1 (mTORC1) activation in the mPFC [127,128].

## 7. Discussion

Several limitations are present in published studies on Rvs. Most studies on inflammation-related diseases have primarily focused on RvD1 [50,63,64,65,66,68,69]. Additionally, studies on the regulation of inflammatory substances and the phagocytosis and efferocytosis of macrophages predominantly employ the RvD series, whereas it is scarcely carried out within the RvE series. Although RvE1 and RvE2 have been demonstrated to increase IL-10 and IL-10R expression, the impact appears limited [73]. Thus, further research is needed to elucidate the specific anti-inflammatory mechanisms of different Rv types.

Additionally, in this review, we noted only the inhibition of inflammatory factors with Rvs, but it has been reported that Rvs are also involved in inhibiting the activity of several transient receptor potential (TRP) channels by regulating G protein-coupled receptors (GPCRs) [129,130]. RvE1 is involved in regulating the activity of TRPV1 by activating ChemR23 or inhibiting BLT1 [130]. ChemR23 and BLT1 are GPCRs that may be co-expressed with TRPV1 in DRG neurons or the spinal cord dorsal horn [131,132]. These two GPCRs have opposite effects. Activation of ChemR23 blocks capsaicin-induced TRPV1 activation, and RvE1 is an agonist of ChemR23 [92,133]. On the other hand, activation of BLT1 induces TRPV1-induced Ca^2+^ influx, and RvE1 is an antagonist of BLT1 [132,134]. The activation of GPR23 by RvD1 or AT-RvD1 inhibits the activation of TRPA1, V3, and V4 [135]. RvD2 has been reported as a potent inhibitor of TRPV1 and A1 through the activation of GPR18 [93]. Further research on ion channel control by Rvs is also required for effective pain control through Rvs.

Moreover, research on the subtype-specific effects of Rvs on pain remains limited. Most studies exploring the analgesic mechanisms have focused on RvD1, RvD2, and RvE1. While Ji et al. recently reported the analgesic effect of RvD5 in a neuropathic pain model, the underlying mechanism responsible for sex-specific differences in RvD5 remains unclear [91].

Further, drugs and mediators that induce inflammatory pain, along with Rvs, were administered to the hind paw to observe an alleviation of inflammatory pain. Subsequently, studies examining the regulation of inflammation by Rvs were primarily conducted within the hind paw region with limited research on the changes in inflammatory factors in crucial areas, such as the DRG or spinal cord.

Finally, the commercialization of Rvs is expected to require not only research on the mechanism of Rv bioactivity but also the development of production process technology. Further research on the mechanism is required to select the appropriate route and method of administration of Rvs, especially considering the prevailing oral approach for administering NSAIDs, which is a simple administration route. This highlights the compelling need to scrutinize the analgesic potential of Rvs via simple administration routes. Moreover, effective supply is also essential for Rvs to be commercialized. Although recombinant LOX-based chemical synthesis technology of some SPMs, such as RvD2 to D5, RvE1, and RvE2, has already been developed, it is currently used only for research due to many limitations in various aspects, such as safety, efficiency, and economy. These limitations pose significant challenges to the practical application of this technology in the commercial production of Rvs.

## 8. Conclusions

Rvs show remarkable efficacy in alleviating both inflammatory and neuropathic pain by regulating immune and glial cells, thereby resolving inflammation in the nervous system. Rvs are endogenous compounds that offer a promising therapeutic potential and an alternative to conventional pain medications, presenting the advantage of reduced side effects commonly associated with current treatments. Continued and comprehensive research is imperative to fully harness the therapeutic potential of Rvs and pave the way for their integration into effective clinical pain management strategies.

## Figures and Tables

**Figure 1 pharmaceuticals-16-01366-f001:**
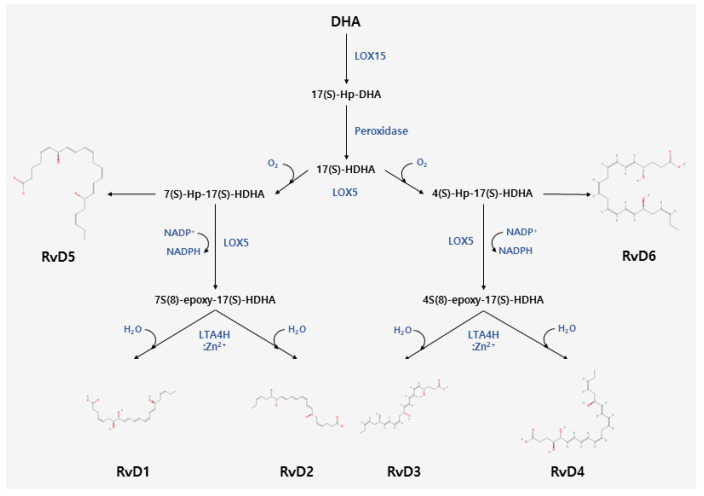
Schematic of D-series resolvins biosynthesis and enzymatic pathway. RvDs are synthesized from DHA via several types of LOX and LTA4H. 17(S)-HDHA converted from DHA is oxidized into two hydroperoxyl intermediates via LOX5, which can be converted into three types of RvDs. RvD1, RvD2, and RvD5 are synthesized from 7(S)-Hp-17(S)-HDHA. RvD3, RvD4, and RvD6 are synthesized from 4(S)-Hp-17(S)-HDHA. 17(S)-HDHA, 17(S)-hydroxy DHA; 4(S)-Hp-17(S)-HDHA, 4(S)-hydroperoxy-17(S)-HDHA; 7(S)-Hp-17(S)-HDHA, 7(S)-hydroperoxy-17(S)-HDHA DHA, docosahexaenoic acid; LOX, lipoxygenase; LTA4H, leukotriene A4 hydrolase; RvDs, D-series Rvs; RvDn, Rv Dn (n = 1~6).

**Figure 2 pharmaceuticals-16-01366-f002:**
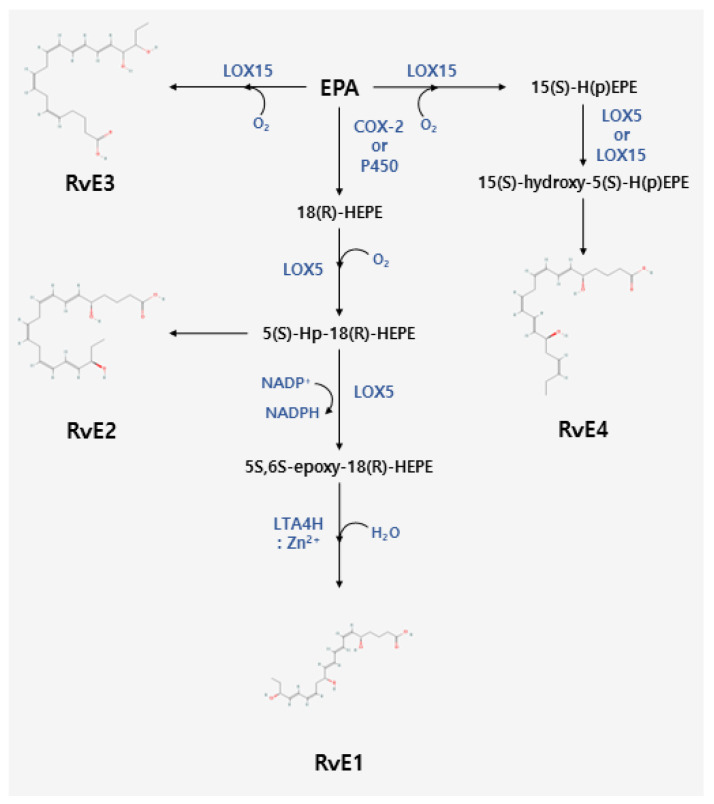
Schematic of E-series resolvins biosynthesis and enzymatic pathway. RvEs are synthesized from EPA. RvE1 and RvE2 are converted from 5(S)-Hp-18(R)-HEPE. EPA may be oxidated to RvE3 or 15(5)-H(p)EPE via LOX15. 15(5)-H(p)EPE is converted into RvE4 through 15(S)-hydroxy-5(S)-H(p)EPE. 18(R)-HEPE, 18(R)-hydroxy EPA; 5(S)-Hp-18(R)-HEPE, 5(S)-hydroperoxy-18(S)-HEPE; 15(S)-H(p)EPE, 15(S)-hydroperoxy EPA; COX-2, cyclooxygenase-2; EPA, eicosapentaenoic acid; RvEs, E-series Rvs; Rv En (n = 1~4).

**Figure 3 pharmaceuticals-16-01366-f003:**
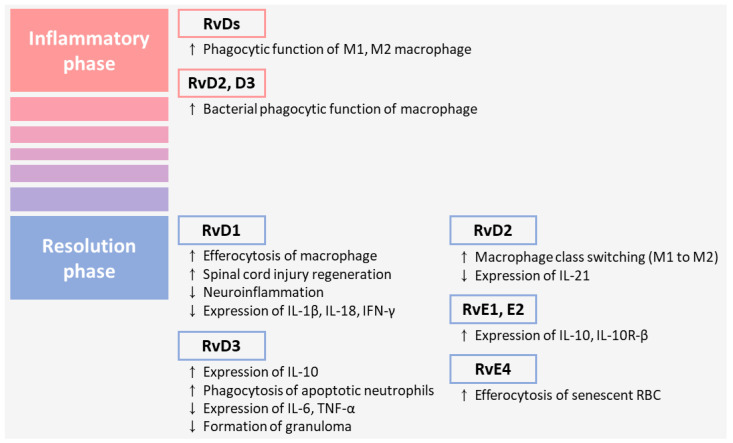
Pro-resolving and inflammatory function of resolvins. Rvs are known to induce the resolution of inflammatory reactions through various mechanisms; they inhibit the expression of pro-inflammatory factors, promote the expression of anti-inflammatory factors, induce class switching of macrophages, restore tissue homeostasis, relieve neuroinflammation, and clear inflammatory immune cells or senescent red blood cells. On the contrary, RvDs promote the phagocytic function of macrophages.

**Table 1 pharmaceuticals-16-01366-t001:** Analgesic effects of resolvins in inflammatory pain models.

Pain Model	Treatment	Pain Behavior	Inflammatory Factor	Tissue	Ref
Formalin- induced pain	RvD5	Spontaneous	-	-	[91]
	RvE1	Spontaneous	-	-	[92]
	RvD2	Spontaneous	-	-	[93]
	RvD1	Spontaneous	-	-	[94]
Capsaicin- induced pain	RvE1	Spontaneous	-	-	[92]
	RvD2, RvE1	Spontaneous	-	-	[93]
CFA- induced pain	RvE1 RvD1	Heat	-	-	[92]
	RvD2	Heat	-	-	[93]
	RVD1	Mechanical, Heat	-	-	[94]
	AT-RvD1	Mechanical	TNF-α, IL-1β↓	Hind paw	[95]
Carrageenan-induced pain	RvE1 RvD1	Heat	TNF-α, IL-1β, IL-6↓	Hind paw	[92]
	RvD2	Mechanical, Heat	-	-	[93]
AITC- induced pain	RvD1, RvD2	Spontaneous	-	-	[93]
IL-17- induced pain	RvD2	Mechanical, Heat	CXCL1↓	Spinal cord	[96]
TNF-α- induced pain	RvE1	Mechanical, Heat	-	-	[92]

Rv treatment in inflammatory pain models alleviates pain responses such as mechanical and thermal hypersensitivity and spontaneous pain. Several results show that increased inflammatory cytokines in the ipsilateral hind paw are alleviated with Rv treatment. AITC, allyl isothiocyanatel; CFA, complete Freund’s adjuvant; CXCL-1, C-X-C motif chemokine ligand 1; IL, interleukin; TNF, tumor necrosis factor.

**Table 2 pharmaceuticals-16-01366-t002:** Analgesic effects and inflammatory factor regulation by resolvin treatments in neuropathic pain models.

Pain Model	Treatment	Pain Behavior	Inflammatory Factor	Tissue	Ref
CIPN mouse	RvD1	Mechanical, Heat, Cold	CD68↓/IL-10↑	DRG, Sciatic nerve	[105]
	RvD5	Mechanical	-	-	[91]
CCI mouse	RvD2	Mechanical, Heat	IL-17, CXCL1, GFAP↓	Spinal cord	[96]
	RvE1	Mechanical	Iba-1, GFAP, TNF-α↓	Spinal cord	[106]
SNL rat	RvE1	Mechanical, Heat	-	-	[106]
	AT-RvD1	Mechanical, Heat	IL-1β, IL-18↓ TNF-α, Iba-1↓	Spinal cord	[107]

Rv treatment in neuropathic pain models alleviates pain responses such as mechanical and thermal hypersensitivity. Decreased inflammatory cytokines are confirmed at central and peripheral nerve system levels. In addition, activation changes in peripheral immune cells and macrophage expression patterns were confirmed with biomarkers in some studies. CCI, chronic constriction injury; CIPN, chemotherapy-induced peripheral neuropathy; CXCL1, C-X-C motif chemokine ligand 1; DRG, dorsal root ganglia; GFAP, glial fibrillary acidic protein; Iba, allograft inflammatory factor; IL, interleukin; SNL, spinal nerve ligation; TNF, tumor necrosis factor.

## Data Availability

Data is contained within the article.

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
