# Peer review of "Role of Resolvins in Inflammatory and Neuropathic Pain"

_pharmaceuticals, 2023, doi:10.3390/ph16101366_

Round 1

Reviewer 1 Report

The present review is a good addition to the literature on inflammatory pain. Traditionally, nonsteroidal anti-inflammatory drugs (NSAIDs), such as cyclooxygenase inhibitors, have been used to manage inflammatory pain. However, the side effects, including gastrointestinal, cardiovascular, hepatic, renal, cerebral, and pulmonary complications, have been reported in multiple placebo-controlled trials and meta-analyses studies. Therefore, the development of effective and safe treatments for inflammatory pain has been studied using various inflammatory pain models.

Authors did a good job in highlighting the role of resolvins in inflammatory and neuropathic pain.  I will suggest authors add a paragraph or two about the preparations of resolvin D and E series that have been used in the discussed studies. How easy is it to make resolvin preparations, their cost compared to available NSAIDs, and upscaling and route of administration.

Line 64. Zn2+..Please use Zn2+ …check it throughout the manuscript.

Minor typos observed, that can be managed by the editorial team.

Reviewer 2 Report

This manuscript represents a good level of the analyzed data. Nevertheless, authors should little revise the text.

Main notes:

1.      The Introduction section and a list of references could be complemented and renewed using the articles that were published recently (for instance, regarding different natural drugs for chronic pain management):

a.       Román-Vargas Y, Porras-Arguello JD, Blandón-Naranjo L, Pérez-Pérez LD, Benjumea DM. Evaluation of the Analgesic Effect of High-Cannabidiol-Content Cannabis Extracts in Different Pain Models by Using Polymeric Micelles as Vehicles. Molecules. 2023 May 24;28(11):4299. doi: 10.3390/molecules28114299. PMID: 37298776; PMCID: PMC10254120.

b.       Santonocito S, Donzella M, Venezia P, Nicolosi G, Mauceri R, Isola G. Orofacial Pain Management: An Overview of the Potential Benefits of Palmitoylethanolamide and Other Natural Agents. Pharmaceutics. 2023 Apr 9;15(4):1193. doi: 10.3390/pharmaceutics15041193. PMID: 37111679; PMCID: PMC10142272.

c.        Ilari S, Proietti S, Russo P, Malafoglia V, Gliozzi M, Maiuolo J, Oppedisano F, Palma E, Tomino C, Fini M, Raffaeli W, Mollace V, Bonassi S, Muscoli C. A Systematic Review and Meta-Analysis on the Role of Nutraceuticals in the Management of Neuropathic Pain in In Vivo Studies. Antioxidants (Basel). 2022 Nov 28;11(12):2361. doi: 10.3390/antiox11122361. PMID: 36552569; PMCID: PMC9774415.

2.      Since the term "resolvins" occurs in the text more than 100 times, it would be appropriate to use an abbreviation for it.

3.      In section 2, it is worth adding some information about natural sources for obtaining resolvins and different ways of their entry into the human body or laboratory animals.

4.      The Conclusions section should be shortened and presented more succinctly.

5.      Lines 276-286 - References to literary sources are not appropriate in the Conclusions of the article, so it is better to move them to the previous sections.

Reviewer 3 Report

This is interesting review about the role of resolvins in inflammatory and neuropathic pain.

The manuscript contains only two Figures. Please present the most important anti-inflammatory effects of resolvins on an additional Figure. Some of these effects are mentioned in subsection 3.

Please mention the following points in the revised manuscript:

1 In the manuscript it is suggested that resolvins may be good alternative for NSAIDs which cause different side effects (line 155, line 296). Do resolvins cause any side effects ?

2 Many drugs which reduce pain inhibit sodium channels. Do resolvins inhibit these channels ?

3 Do resolvins affect functioning of the brain ?

Moderate editing of English language required.
